# Dual phase patterning during a congruent grain boundary phase transition in elemental copper

Lena Langenohl [1,4], Tobias Brink [1,4✉], Rodrigo Freitas[2], Timofey Frolov [3✉], Gerhard Dehm [1✉] & Christian H. Liebscher [1✉]

The phase behavior of grain boundaries can have a strong influence on interfacial properties. Little is known about the emergence of grain boundary phases in elemental metal systems and how they transform. Here, we observe the nanoscale patterning of a grain boundary by two alternating grain boundary phases with distinct atomic structures in elemental copper by atomic resolution imaging. The same grain boundary phases are found by computational grain boundary structure search indicating a first-order transformation. Finite temperature atomistic simulations reveal a congruent, diffusionless transition between these phases under ambient pressure. The patterning of the grain boundary at room temperature is dominated by the grain boundary phase junctions separating the phase segments. Our analysis suggests that the reduced mobility of the phase junctions at low temperatures kinetically limits the transformation, but repulsive elastic interactions between them and disconnections could additionally stabilize the pattern formation.

[1] Max-Planck-Institut für Eisenforschung GmbH, Max-Planck-Straße 1, 40237 Düsseldorf, Germany. [2] Department of Materials Science and Engineering, Massachusetts Institute of Technology, Cambridge, MA 02139, USA. [3] Lawrence Livermore National Laboratory, Livermore, CA 94550, USA. [4]These authors contributed equally: Lena Langenohl, Tobias Brink. ✉email: t.brink@mpie.de; frolov2@llnl.gov; dehm@mpie.de; liebscher@mpie.de

Grain boundaries (GBs) are the interfaces separating adjoining crystallites and can impact the mechanical[1–4] and electronic[5–8] properties of polycrystalline materials. GBs can exist in multiple stable and metastable states, which are typically associated with differences in the atomic structure of the GB core, and it was proposed that they can undergo phase transitions[9–14]. The terms GB phase[13] or complexion[11,12,14] have been introduced as analogs to bulk phases to underline that these interface phases can only exist in contact to other bulk phases. Each GB phase is characterized by distinct thermodynamic excess properties[9,10,15], which can have an impact on, for example, sliding resistance[16], GB migration[17], or shear-coupled GB motion[18]. In metallic systems, most experimental evidence for GB phase transitions is inferred indirectly from abrupt changes in diffusivity[19–21] or GB migration[22]. One of the first direct observations of two different structures within one GB was obtained for NiO[23].

Experimental evidence of congruent GB phase transitions in elemental metals is lacking, since they are difficult to observe. A congruent GB phase transition is characterized by transformations limited to the GB core without a change in grain misorientation and GB plane[12]. These transitions have mostly been studied using atomistic modeling of [001] tilt GBs in fcc copper[15,24,25], in various tungsten GBs[26], and magnesium[27]. Recently, two different GB phases were observed experimentally in Σ19b ⟨111⟩ {1 7 8} GBs in copper by atomic resolution scanning transmission electron microscopy (STEM)[28]. Using atomistic simulations, it was found that only one GB phase was stable over the temperature range from 0 to 800 K at ambient pressure and a congruent phase transition would only be possible by applying tensile or shear stresses[28]. The room temperature observations were related to stresses stabilizing the metastable GB phase and a reduced mobility of the GB phase junction at low temperatures, kinetically trapping the metastable phase.

Grain boundary phase junctions themselves, which are line defects separating two GB phases[13], therefore play an important role in the energetics and kinetics of GB phase transitions. The junctions have a dislocation character, similar to disconnections[29,30], but their Burgers vectors include contributions from the structural difference of the abutting GB phases[31]. Since they interact elastically via stress fields in a similar manner as dislocations do[30], it is expected that they significantly contribute to the nucleation barrier of GB phases and interact with other GB defects[31]. However, their character and related influence on GB phase transitions has barely been studied[32].

In the present work, we investigate diffusionless, congruent GB phase transitions in Σ37c ⟨111⟩ {1 10 11} GBs by atomic-resolution STEM and atomistic modeling. The misorientation between both grains changed by only 4° compared to an earlier study of GB phases in a Σ19b GB[28]. Even though similar GB phases occur, their thermodynamic stability is markedly different: Here, one GB phase is stable in low-temperature regimes (below 460 K, called the domino phase) whereas the other GB phase (called pearl phase) is energetically favored at elevated temperatures. The GB phase transformation can thus occur under ambient pressure by temperature alone. We further discuss the structure and properties of these GB phases and the influence of GB defects and phase junctions on an experimentally observed GB phase pattering.

## Results

### Experimental observation of GB phases. A 1 μm thick ⟨111⟩ epitaxially grown Cu thin film was used as a template material (see Supplementary Fig. 1 for an inverse pole figure map of the film). Plane-view focused ion beam (FIB) lamellas of Σ37c ⟨111⟩

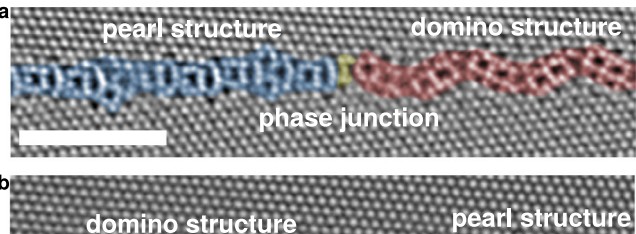

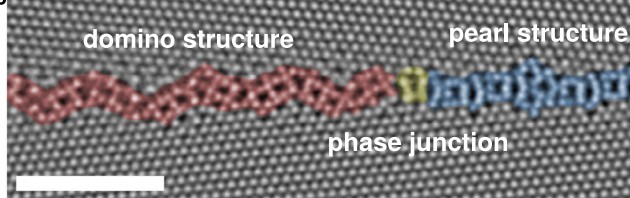

**Fig. 1 HAADF-STEM images of two GB phases separated by phase junctions in a nearly symmetric Σ37c ⟨111⟩ {1 10 11} GB.** The GB plane remains unchanged by the phase transition and no faceting is observed. **a**, **b** Shows these junctions at two different positions on the same film. The scale bars represent 2 nm.

GBs were lifted out from two different positions in the film and their atomic structures were investigated by high-angle annular dark-field (HAADF) STEM. Two different GB structures could be observed, both of them occurring frequently. Figure 1 shows the structures and the phase junctions between them at two different positions along a nearly symmetric GB. We termed these distinct GB phases pearl (blue) and domino (red) due to their similarity to the structures in Σ19b ⟨111⟩ {178} GBs[28]. The misorientation between both grains is 50.0(3)°, which is within the Brandon criterion[33] for a Σ37c GB (nominal misorientation angle of 50.6°).

Two different overview montages consisting of multiple individual HAADF-STEM images encompassing a total length of up to 300 nm of the GB are shown in Fig. 2 and Supplementary Fig. 2 (version with full resolution). The GB adopts a slight curvature and deviates in some regions from the symmetric orientation. Interestingly, the GB is composed of alternating pearl and domino segments. Taking into account only fragments deviating <5° from the symmetric case, the domino GB phase comprises 77% of the GB with segment lengths ranging from 40 nm to more than 100 nm. The pearl GB phase segments adopt lengths between 10 and 25 nm, taking up a total fraction of about 23%. These observations suggest that the domino phase is more stable at low temperatures, but a large amount of remaining pearl phase is surprising: At constant stress, the phase coexistence region for congruent GB phase transitions of elemental systems is restricted to a single temperature, thereby practically excluding thermodynamically stable coexistence[14]. To understand the observed GB phase patterning by two structurally distinct GB phases, we first explore their thermodynamic excess properties and defects in detail. This includes the phase junction and the strain field of possible disconnections compensating the slightly asymmetric orientation of the GB segments. In addition, the influence of the phase junction kinetics is considered.

**Structure and properties of the grain boundary phases.** Atomic resolution HAADF-STEM images of both pearl and domino GB phases are shown in Fig. 3a, c. Furthermore, both grain boundary structures shown in Fig. 3b, d were obtained by molecular statics simulations at 0 K using an embedded-atom method (EAM) potential[34]. Here, two fcc half-crystals were joined and different relative displacements were sampled following the γ-surface method until the experimentally observed structures were obtained.

The domino phase consists of domino I and II motifs, which can be mapped onto each other by a 180° rotation around the

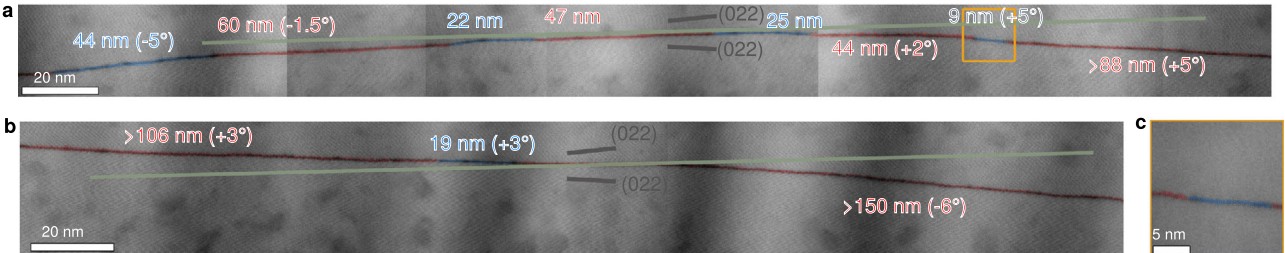

**Fig. 2 Overviews of two, more than 300 nm long, GB segments assembled from multiple HAADF-STEM images of near-symmetric areas of a Σ37c ⟨111⟩ GB. a, b** The GBs consist of multiple, alternating domino (red) and pearl (blue) segments. The lengths of each segment are indicated as well as the deviation off the symmetric GB plane (green line). **c** Magnified view of the orange region in (**a**). The domino segments in symmetric areas (<5° off the symmetric case) are between 40 and 60 nm long, whereas the pearl segments are shorter with 10–40 nm. A high-resolution version of this image can be found in Supplementary Fig. 2.

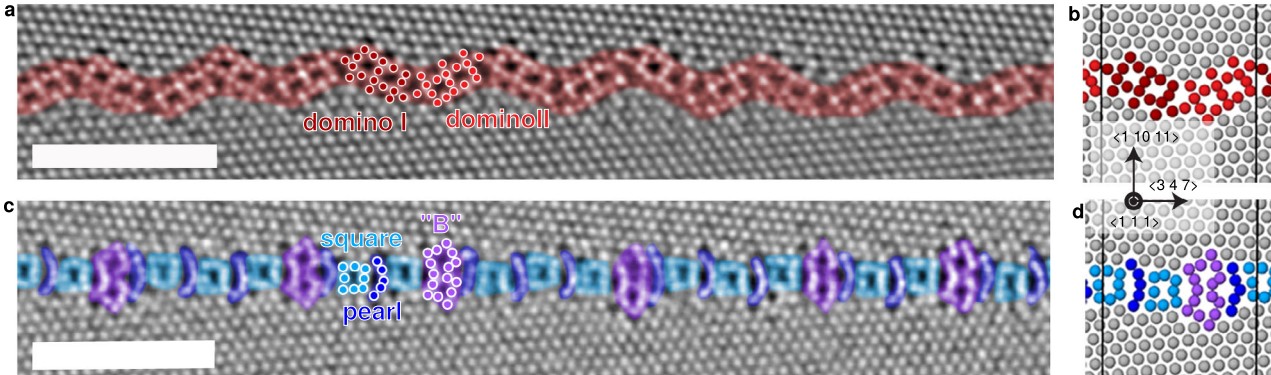

**Fig. 3 The two GB phases in a nearly symmetric Σ37c ⟨111⟩ {1 10 11} GB in Cu as observed by HAADF-STEM.** The sub-atomic structures of the domino phase (**a**) and the pearl phase (**c**) are indicated by the colouring. The scale bars represent 2 nm. **b, d** Corresponding structures obtained from atomistic simulations with the EAM potential by Mishin et al.[34]. The motifs are the same as in the experiment. The black lines indicate the unit cell of the GB structure.

⟨347⟩ direction parallel to the GB. In structural unit notation[35], we denote the domino unit cell as |D ⊓|. Disconnections take the form of extending or shortening one of the D motifs by a square or change the arrangement of both D motifs as highlighted in Supplementary Fig. 11. The pearl phase is more complex and consists of square motifs (S), either connected by a pearl chain (P) or a B unit after two or three repetitions. The structural unit obtained by the $\gamma$-surface method (Fig. 3d) has a |S P ƨ B P| unit cell, indicating that the varying distance between B units in the experiment represents defects, which possibly compensate the slight asymmetry of the GB and slightly smaller misorientation angle between both grains (50.0° instead of the perfect 50.6°).

In addition to matching the experimental structures, we also sampled the phase space of possible GB structures at $T = 0$ K efficiently—even for structures that have an excess number of atoms per unit cell—using an evolutionary algorithm[25] implemented in the USPEX code[37,38]. The thermodynamic excess properties of all structures are shown in Fig. 4a–d and Supplementary Fig. 3. These include the grain boundary energy $\gamma_0$ at 0 K, the excess volume $[V]$, the excess stresses $[\tau_{ij}]$, the excess number of atoms $[n]$ (see "Methods" for its definition), and the excess shear $[\mathbf{B}]$. The latter corresponds to the microscopic translation vector between the two crystallites when no external stress is applied to the system. The notation $[Z]$ refers to the excess of property $Z$ in a system with a grain boundary over a perfect crystal with the same number of atoms[36]. In order to obtain intensive values for all excess properties, $\gamma$, $[V]$, and $[\tau_{ij}]$ are normalized by the grain boundary area.

Using $k$-means clustering with $k = 2$ on the full dataset of all structures with $\gamma_0 < 0.95$ J m$^{-2}$, we can cleanly separate the structures into pearl-like and domino-like phases and their defective variants. This can be visualized in pair plots, where two different properties are plotted against each other (Fig. 4a–d and Supplementary Fig. 3). The excess stress component $[\tau_{22}]$ is the best single predictor separating the two phases. The domino structure has the lowest grain boundary energy and represents the ground state at 0 K. The lowest-energy phase in the pearl cluster, indicated with an orange diamond symbol in Fig. 4 and termed pearl #1, does not resemble the experimentally observed one. The B motif is replaced by an Ω motif: |P S P ƨ Ω|. Further manual search revealed a pearl #2 variant with 6 mJ m$^{-2}$ higher grain boundary energy and the experimentally observed B motif (blue triangle in Fig. 4). The excess shears $[\mathbf{B}]$ differ by $(0.006, 0.105, 0.131)$ Å between the two variants and the other excess properties are similarly close. The properties of the defect-free domino, pearl #1, and pearl #2 structures are listed in Table 1. There are several intermediary structures in between pearl #1 and #2 (indicated by the connecting line in Fig. 4d and explored in detail in Supplementary Fig. 4), which suggests that these states represent different microstates of the pearl phase at elevated temperatures. From STEM image simulations on pure pearl #1, pearl #2, and a mixture of both stacked in ⟨111⟩ direction (see Supplementary Fig. 7), we can conclude that the mixture would not be distinguishable from a pure pearl #2 structure in experimental HAADF-STEM images.

Finally, a change in the excess number of atoms $[n]$ has been connected to diffusion-driven grain-boundary phase transitions

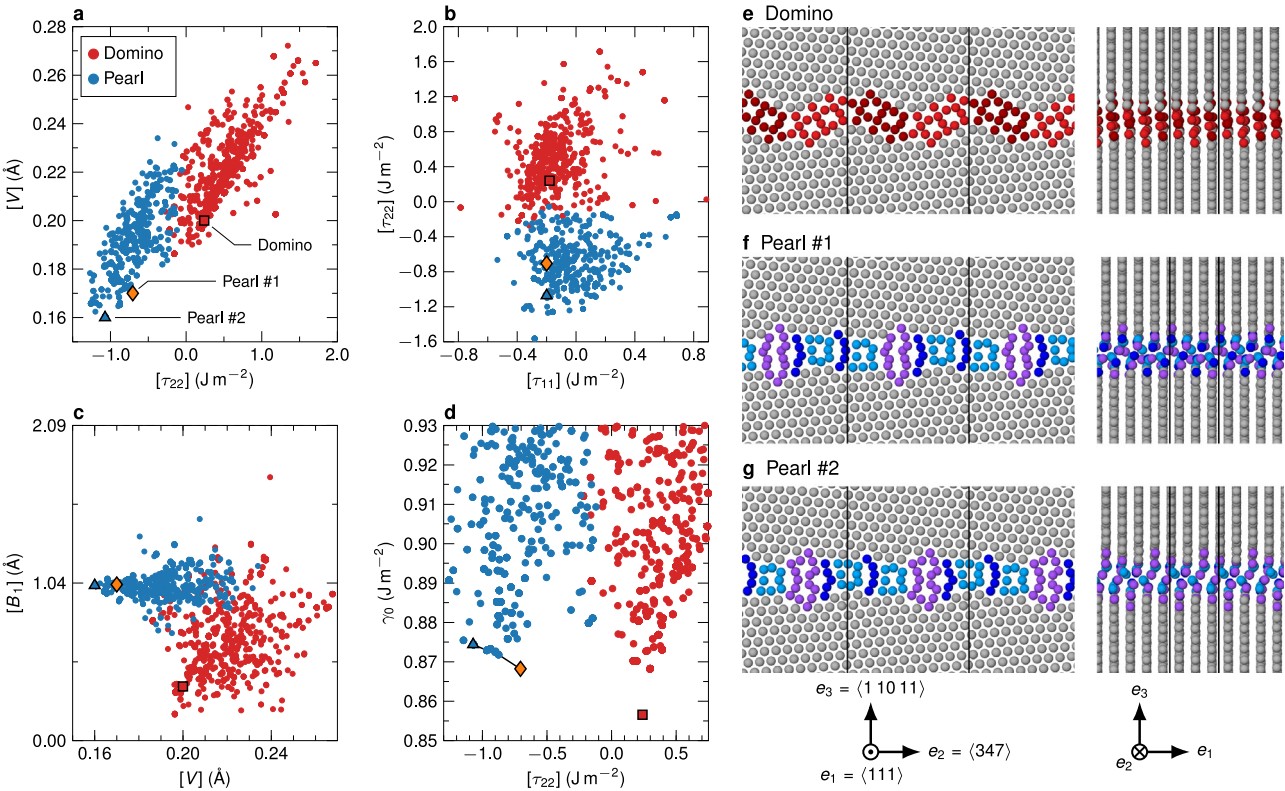

**Fig. 4 Grain boundary phases predicted by the EAM potential discovered with phase-space sampling by an evolutionary algorithm. a–c** Pair plots of different GB excess properties that make the separation into two main clusters of data points visible. The color coding is according to a clustering algorithm that takes several excess properties into account. Many of the structures that were discovered are simply defective, i.e., they contain disconnections or other defects. Three defect-free base structures are highlighted by the triangle, diamond, and square symbols, in which two are microstates of the pearl phase (the triangle and diamond) and one refers to the domino phase (the square). The best predictor to separate the clusters is $[\tau_{22}]$, but the excess shear also provides a good indicator. **d** When plotting the GB energy over this predictor, a clear separation between low-energy domino-like and pearl-like structures can be seen. The low-energy pearl variants consist of either pearl #1 or #2, or a mixture of the two (indicated by the line connecting the data points and explored in more detail in Supplementary Fig. 4). **e–g** Snapshots of the low-energy structures from two directions. The unit cells are marked by black lines.

**Table 1 Excess properties of the three low-energy GB structures in the $\Sigma 37c$ $\langle 111 \rangle$ $\{1\,10\,11\}$ GB as predicted by the computer model.**

|  | domino | pearl #1 | pearl #2 |  |
|---|---|---|---|---|
| $\gamma_0$ | 0.857 | 0.868 | 0.874 | J m$^{-2}$ |
| $[V]$ | 0.200 | 0.170 | 0.160 | Å |
| $[\tau_{11}]$ | −0.18 | −0.20 | −0.20 | J m$^{-2}$ |
| $[\tau_{22}]$ | 0.24 | −0.71 | −1.07 | J m$^{-2}$ |
| $[\tau_{12}]$ | 0.0 | ±0.25 | ±0.07 | J m$^{-2}$ |
| $[B_1]$ | 0.359 | 1.034 | 1.028 | Å |
| $[B_2]$ | 0.000 | ±0.105 | ±0.210 | Å |
| $[n]$ | 0 | 0 | 0 |  |

We follow the conventions used by Frolov and Mishin[36], where index 1 corresponds to the tilt axis $\langle 111 \rangle$, 2 corresponds to the $(3\,4\,7)$ direction parallel to the GB, and 3 corresponds to the grain boundary normal $(1\,10\,11)$. The signs of $[\tau_{12}]$ and $[B]$ depend on the choice of a specific orientation of the phase and the coordinate system (see Supplementary Fig. 5 for details). $[B_3]$ is equal to $[V]$ when $[n] = 0$.

in various metals[15,25,26,39]. However, in the present case, all ground-state structures of the defect-free GB phases adopt values of $[n] = 0$ (see Table 1), indicating that the GB phase transition is not driven by the insertion or removal of atoms. In any case, where two GB phases have the same value of $[n]$, a phase transition can occur by local rearrangements of atoms and is thus considered to be diffusionless. In the present GBs, interstitial- or

vacancy-type defects do not lead to different GB phases, but only to defective microstates (Supplementary Figs. 3c and 4).

**Diffusionless grain boundary phase transition.** In a first step to explore the underlying mechanisms leading to the experimentally observed patterning of the GB phases, we calculated their excess free energies to determine GB phase stability and phase transition temperatures. We used the quasi-harmonic approximation[40,41] on the defect-free structures (Fig. 5) and confirmed the results using thermodynamic integration[41,42] (see "Methods" and Supplementary Fig. 6). Figure 5 shows that this system exhibits a GB phase transition temperature of around 460 K = $0.33 T_m$ under constant, ambient pressure, with $T_m = 1358$ K being the melting point of copper. Domino is the stable GB phase at low temperature, and pearl #2 at high temperature. This indicates that the pearl phase is a structure with higher entropy. We estimated the excess entropy of the GB phases via[36,41,43] $[S] \approx -\mathrm{d}\gamma(T)/\mathrm{d}T$. At $T = 0$ K, we obtain excess entropies of ~0.21 mJ m$^{-2}$ K$^{-1}$ for domino and 0.24 mJ m$^{-2}$ K$^{-1}$ for both pearl structures, confirming the higher entropy of pearl.

We used annealing simulations to test the prediction of the phase transition temperature and to obtain partially transitioned systems containing GB phase junctions. The nucleation and phase transition is expected to be quite slow on MD timescales, so we started by annealing a sample containing the domino phase (Fig. 6a) at 800 K. This sample had open surfaces in $\langle 1\,10\,11 \rangle$ direction, but was otherwise periodic so that the grain boundary

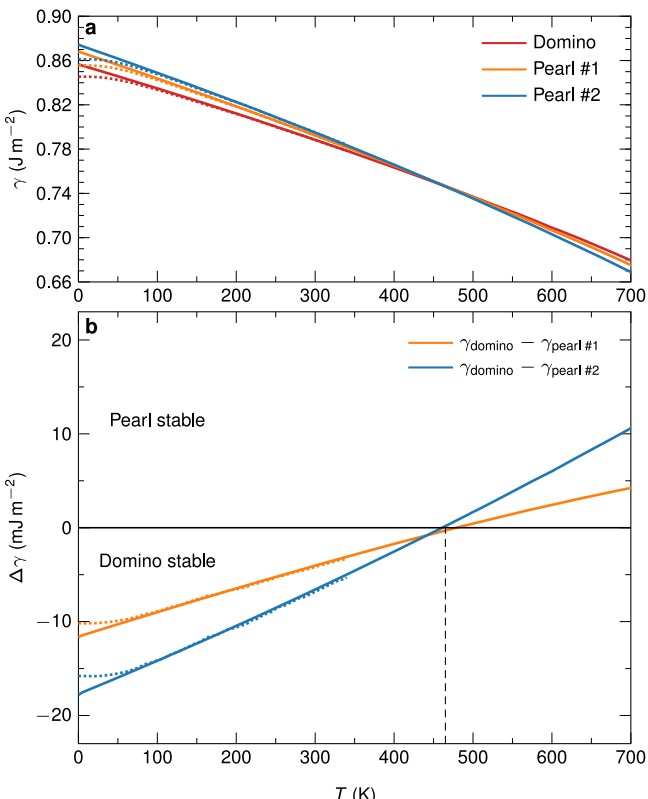

**Fig. 5 Free energy calculation with the EAM potential using the quasi-harmonic approximation.** **a** Plot of the GB free energies $\gamma$ for the three low-energy structures we discovered. The dashed lines indicate an approximation with quantum-mechanical effects, the solid lines represent a purely classical approximation. The classical approximation yields equivalent results above 100 K. **b** The free energy differences show that the domino phase is stable below ~460 K, while pearl #2 is stable above that temperature.

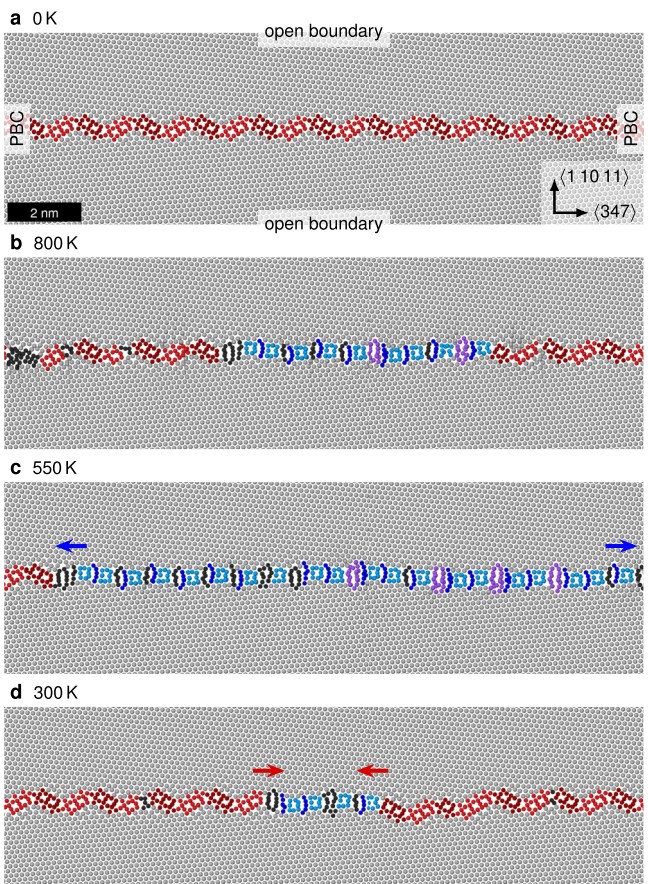

**Fig. 6 Phase transition simulations.** **a** We start from the domino structure with periodic boundary conditions (PBC) in the GB plane and open boundaries normal to it. **b** The pearl phase nucleates at 800 K and **c** keeps growing after reducing the temperature to 550 K (snapshot after 4 ns). The pearl structures contain many defects (disconnections, black). For a slowly cooled pearl sample with less defects see Supplementary Fig. 7. **d** Regrowth of the domino phase at 300 K observed after 2 ns for a system thickness of three atomic layers in tilt direction. All pearl phase disappeared after an additional 0.2 ns. The initial state was extracted from (**b**). All images are slices of width 1 nm.

had no contact to the open surfaces and nucleation was homogeneous. Its width in tilt axis direction was 6.3 nm, corresponding to 30 {111} layers. After around 1.3 ns, the pearl phase began to nucleate and grew from within the parent domino phase (Fig. 6b) indicating that homogeneous nucleation is possible, likely because the phase transition requires no diffusion. This fits to the experimental observation of multiple nanoscale segments of the pearl phase occurring within the GB (Fig. 2), instead of nucleation only at e.g. GB triple junctions. Thus, multiple nucleation sites for the pearl phase seem to be easily accessible within the domino structure. We took the sample from Fig. 6b and annealed it further at 550 K, which lead to a growth of the pearl phase segment since it is above the transition temperature of 460 K (Fig. 6c). Below the transition temperature, the migration of the GB phase junction is too slow to be observed in MD timescales. We could accelerate the process by reducing the system to a thickness of three atomic {111} layers, in which case the pearl phase was observed to dissolve at 300 K (Fig. 6d). A similar dependence of phase junction mobility on system thickness was observed before[28]. An analysis of displacement vectors during the GB phase transition confirmed that only local atomic shuffling is required to create a pearl nucleus in the domino phase, showing that this GB phase transition is diffusionless (Supplementary Fig. 8).

The high temperature pearl phase was observed to consist of a mixture of both pearl #1 and #2 variants, but distinguishing them is difficult due to the large amount of defects, especially in between the S motifs. A cleaner pearl GB phase could be obtained by further annealing at 800 K and subsequent cooling (Supplementary Fig. 7). This phase contains both the B and $\Omega$ structural units of the two pearl variants stacked in $\langle 111 \rangle$ direction, indicating that pearl #1 and #2 resemble microstates of a combined pearl phase, at least at high temperatures in the model.

Finally, we excluded that diffusion could lead to the appearance of other new phases by annealing a pearl grain boundary at 800 K for 30 ns with open surfaces in $\langle 347 \rangle$ direction. It has been shown in previous work[15] that such boundary conditions are conducive to GB phase transitions, but no novel phase appeared in our simulations, supporting the conclusion that pearl is the stable phase at high temperatures.

**Grain boundary phase junction.** The GB phase junction is a 1D line defect that separates both the pearl and domino GB phases[13]. Besides being important for the kinetics of the GB phase transition, it plays a vital role in the nucleation of GB phases and the coalescence of GB phase segments. It was recently established that the phase junction is characterized by a Burgers vector, which depends on the excess properties of the abutting GB phases[31]. In a single GB phase, line defects have disconnection character and

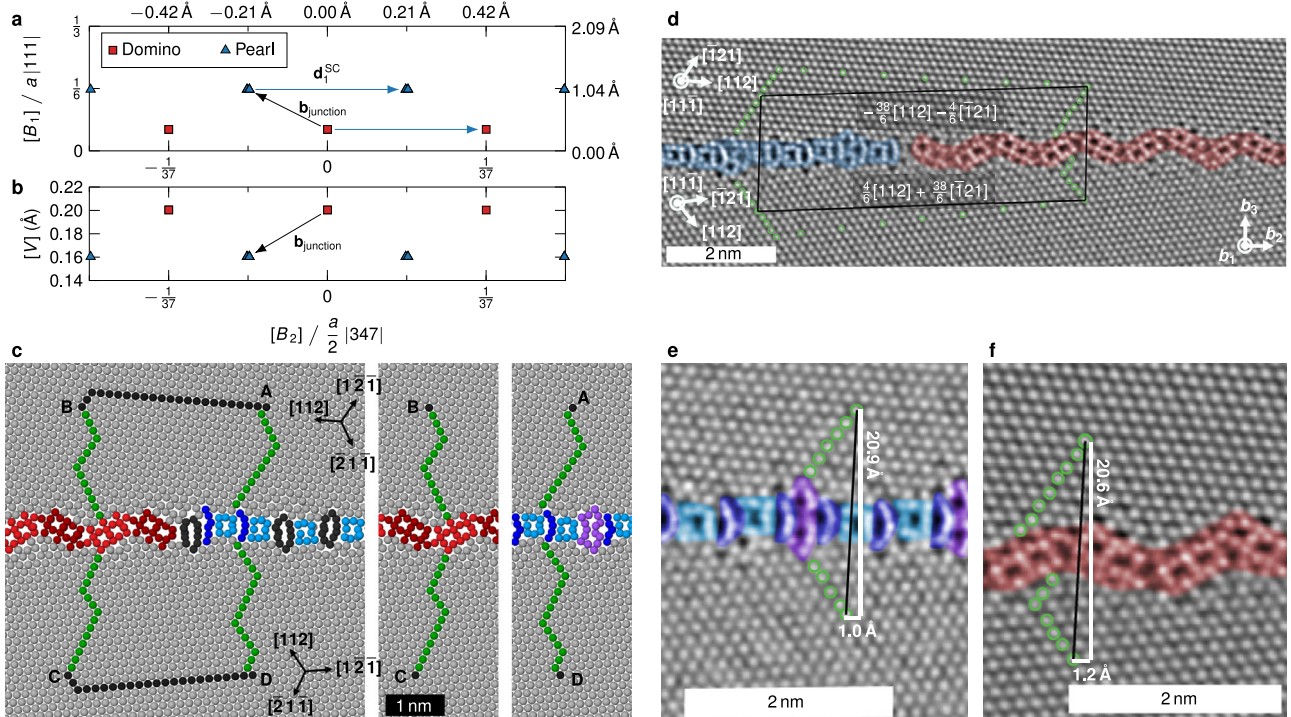

**Fig. 7 Burgers vector of the phase junctions. a, b** Microscopic translations [**B**] between the crystallites for the different phases. The differences in [**B**] correspond to the Burgers vector $b_{junction}$. Different possible [**B**] vectors are equivalent, since they are connected by DSC vectors $d^{SC}$. **c** Burgers circuit on an actual junction in the simulation. The black lines were chosen to be parallel, so that $\overline{AB} + \overline{CD} = \mathbf{0}$. The green lines of the circuit were measured in ideal unit cells of the GB phases to avoid elastic distortions near the junction. **d** Burgers circuit on the experimental image from Fig. 1a using the same method. **e, f** The lines crossing the GB were translated to reference segments far away from phase junctions before measuring in order to reduce elastic distortions. The same images as in Fig. 3 were used. The components parallel and normal to the GB were measured to be $b_\parallel = -0.28$ to $-0.45$ Å and $b_\perp = -0.1$ to $+0.14$ Å. For more circuits at different phase junctions see Supplementary Fig. 11.

their Burgers vectors are also displacement shift complete (DSC) vectors[30]. This is not necessarily the case for phase junctions, where the smallest possible Burgers vector is the difference in the translation vector [**B**] of the abutting GB phases[31]. Nevertheless, the phase junction can also absorb disconnections and we therefore started by finding the dichromatic pattern and DSC vectors of the Σ37c GB (Supplementary Figs. 9 and 10). We then explored the possible [**B**] vectors of the GB phases by constructing bicrystals with different values of $[B_1]$ and $[B_2]$ and running molecular statics ($[B_3]$ is equal to the grain boundary excess volume for $[n] = 0$ and does not require systematic search). We found that apart from the values listed in Table 1, $[B_2]$ can take any value obtained by adding an integer multiple of, e.g., the DSC vector $\mathbf{d}_1^{SC} = (0, a/74 \cdot |347|, 0) \approx (0, 0.420 \text{ Å}, 0)$, as expected (Fig. 7a, b). The resulting Burgers vector of the phase junctions separating the GB phases without additional disconnection content is shown in Fig. 7a, b and has a value of

$$\mathbf{b}_{junction} = [\mathbf{B}_{pearl}] - [\mathbf{B}_{domino}]$$

$$= \left(0.669 \text{ Å}, \pm \frac{a}{148}|347|, -0.04 \text{ Å}\right) \quad (1)$$

$$\approx (0.669, \pm 0.210, -0.04) \text{ Å}.$$

In addition, we cooled the simulation from Fig. 6c to 200 K and minimized the structure with regard to the potential energy. We then constructed a Burgers circuit (Fig. 7c) around one of the junctions. The circuit was chosen to contain two parallel, equally long lines (black) that add to zero. The Burgers vector can then be calculated as $\mathbf{b}_{junction} = \overline{BC} + \overline{DA}$, where the two lines across the GB are measured in single-phase simulation cells to avoid elastic

distortion due to the junction and the defects in the pearl phase (see "Methods" for details on the procedure[31]). We obtained a value of $(0.672, -0.205, -0.039)$ Å, which corresponds to the predicted value within the expected accuracy of the atomic positions in the simulation.

We investigated the Burgers vector of the phase junction experimentally using the same method[31]. A complete Burgers circuit is drawn counter-clockwise around the phase junction as shown in Fig. 7d. The lines crossing the GB were measured in reference images far from the junction. As we observe a projection of the phase junction by HAADF-STEM, only the second and third component, parallel and normal to the GB plane, can be deduced. We calculated a Burgers vector for in total three different phase junctions (see Fig. 7 and Supplementary Fig. 11). The values of the Burgers vector at the phase junction shown in Fig. 7d ($b_a$) and the ones shown in Supplementary Fig. 11 ($b_b$) and b ($b_c$) have values of

$$\mathbf{b}_a = (?, -0.28 \text{ to} - 0.45 \text{ Å}, +0.14 \text{ to} - 0.1 \text{ Å}) \quad (2)$$

$$\mathbf{b}_b = (?, +0.33 \text{ to} + 0.48 \text{ Å}, +0.05 \text{ to} + 0.2 \text{ Å}) \quad (3)$$

$$\mathbf{b}_c = (?, +0.25 \text{ to} + 0.38 \text{ Å}, +0.05 \text{ to} + 0.26 \text{ Å}) \quad (4)$$

depending on the Burgers circuit drawn around the phase junction. The uncertainty of each measurement is about ±0.2 Å, considering possible sources of error in the experiment, such as small localized residual stresses in the undeformed reference state and non-linear scan distortions leading to sub-Ångström variations in the positions of atomic columns. Thus, the values are in good agreement with $\mathbf{b}_{junction}$ obtained from the computer model. The experimental determination of the Burgers vector of

the GB phase junction may be further complicated by additional disconnections next to or within the phase junction. The smallest possible disconnections in a Σ37c system are shown in Supplementary Fig. 10a, having a minimum length of 0.21 Å parallel to the GB, the $b_2$ component, and 0.12 Å normal to the GB, the $b_3$ component. These smallest disconnections could be added to or subtracted from the experimentally obtained values of the phase junctions' Burgers vector and the value would still be within the range determined from atomistic simulations.

The component of the Burgers vector along the tilt axis of the GB and hence the line sense of the phase junction, $b_1 = 0.67$ Å, is the largest component. Thus the phase junction predominantly adopts a screw-type character. This is consistent with the observations for Σ19b GBs[28], where the phase junction between both GB phases was investigated qualitatively without further calculations. The absolute quantitative evaluation of the Burgers vector is needed to calculate the elastic interactions between GB phase junctions and disconnections. As we will see in the next section, this elastic interaction is mainly responsible for the patterning of both GB phases observed in our experiments at room temperature.

**Mechanisms leading to grain boundary phase patterning.** We now discuss the mechanisms leading to the experimentally observed patterning of the GB by both pearl and domino phases. When the sample is cooled down to room temperature from above 460 K, the phase transition initiates by nucleation of the domino phase within the pearl phase. Similar to bulk phase transitions, nuclei of the domino phase appear due to random thermal fluctuations and will most likely start at the surfaces or interfaces, which lower the energy needed to initiate the process. Detailed investigations of a homogeneous nucleation of GB phases were recently published by Winter et al.[32]. The free energy change during the formation of a nucleus can be written as the sum of the free energy reduction due to the transition to the thermodynamically stable phase and the energy cost of the phase boundary. The phase junction's contribution consists of an elastic interaction energy and a core energy. The core energy of a phase junction of a Σ29 (520) [001] symmetric tilt GB in tungsten was calculated to be extremely anisotropic, being four times lower in the tilt direction than in its normal direction in the GB plane[32]. To our knowledge, that is the only reported value of a GB phase junction core energy. However, in a Σ5⟨100⟩(210) GB in copper[13], a GB phase nucleus was observed to be highly anisotropic as well, also showing an elongated shape in the tilt direction. Thus we assume that the nucleus in the present case has an approximately oval shape and rapidly expands along the ⟨111⟩ tilt axis due to an anisotropic core energy. This implies that the nucleating domino phase becomes multiple times longer in the ⟨111⟩ direction than the ⟨347⟩ one, matching the observed pattern with segment lengths much shorter (10 to 100 nm) than the film thickness (1 μm). We note here that the present investigation is limited to a rather two-dimensional picture and it remains currently unclear whether such a striped GB phase pattern would change, for example, into maze-like patterns in a large-grained bulk polycrystal. In order to observe the 3D arrangement of GB phases within the GB plane, new imaging modalities in the TEM would need to be developed.

The free energy change due to a newly nucleated GB phase can then be calculated by assuming it is enclosed by two parallel phase junction lines by the expression

$$\frac{\Delta G_{\text{nucl}}(T)}{t} = l\Delta\gamma(T) + E_{\text{dip}}, \tag{5}$$

where $l$ is the length of the newly formed phase segment, $t$ is the film thickness, $\Delta\gamma(T)$ is the temperature dependent free energy

difference of the GB phases, and $E_{\text{dip}}$ is the energy of the phase junction dipole per unit line segment, which we assume to be temperature independent, consisting of the core energy and the elastic interaction energy $E_{\text{inter}}$. The elastic interaction energy was first described by Nabarro for two dislocations[44] and later adapted for disconnections[30]. We assume that it is also valid for phase junctions, which have a dislocation content[31]

$$\frac{E_{\text{inter}}}{t} = -\frac{\mu}{2\pi}\ln\frac{l}{\delta_0}\left[(\mathbf{b}^a\cdot\hat{\mathbf{o}})(\mathbf{b}^b\cdot\hat{\mathbf{o}}) + \frac{(\mathbf{b}^a\times\hat{\mathbf{o}})\cdot(\mathbf{b}^b\times\hat{\mathbf{o}})}{1-\nu}\right] - \frac{\mu}{2\pi}\frac{(\mathbf{b}^a\cdot\hat{\mathbf{n}})(\mathbf{b}^b\cdot\hat{\mathbf{n}})}{1-\nu}. \tag{6}$$

Here, $\mathbf{b}^a$ and $\mathbf{b}^b$ are the Burgers vectors, $\mu$ is the shear modulus ($\mu_{\text{Cu}} = 34$ GPa), $\hat{\mathbf{n}}$ the unit vector of the GB plane normal, $\hat{\mathbf{o}}$ the unit line vector (equal to the tilt axis in our case), $\nu$ the Poisson's ratio ($\nu_{\text{Cu}} \approx 0.34$), and $\delta_0$ the core size. We can simplify the equation by choosing our coordinate system such that $\hat{\mathbf{n}} = (0, 0, 1)$ and $\hat{\mathbf{o}} = (1, 0, 0)$:

$$\frac{E_{\text{inter}}}{t} = -\frac{\mu}{2\pi}\ln\frac{l}{\delta_0}\left[b_1^a b_1^b + \frac{b_2^a b_2^b + b_3^a b_3^b}{1-\nu}\right]$$
$$- \frac{\mu}{2\pi}\frac{b_3^a b_3^b}{1-\nu}. \tag{7}$$

Since the core size $\delta_0$ leads to a constant energy contribution independent of the junction distance $l$, we can treat it as an effective core size that already includes $2E_{\text{core}}$ and define $E_{\text{dip}} = E_{\text{inter}}/t$ for simplification.

Once the oval nucleus expands over the whole film thickness, the anisotropy of the core energy can be neglected and the phase junctions can be simplified by a pair of two dislocation lines having opposite Burgers vectors ($\mathbf{b}^a = -\mathbf{b}^b$) and thus an attractive interaction during growth.

The growth of the domino phase inclusions can be limited by the migration of the GB phase junction and the interaction of phase junctions when two neighboring domains grow towards each other. The motion of the phase junction is strongly temperature dependent and may contribute to a stagnation in growth of the domino phase domains below a temperature of 400 K, as was also observed previously[28]. One reason for the reduced mobility could be the large screw ($b_1$) component of the Burgers vector[45]. The role of interacting phase junctions during GB phase coalescence is far less understood. In a perfect GB, it is reasonable to assume that two newly nucleated domino phases have the same translation vector [**B**] and the two nuclei are thus delimited by junctions with the same Burgers vector ($\mathbf{b}_{\text{junction}} = \Delta[\mathbf{B}]$). This means that the two closest junctions have opposite Burgers vectors and attract each other, which would promote the coalescence of domino phase domains. It is conceivable that the [$B_2$] component of the two domino phases is different, but this does not lead to the repulsion of the junctions due to the large $b_1$ component of their Burgers vectors. In this scenario, a patterning is not expected, which is in line with the fact that it was also not observed in the simulations.

However, so far we did not consider the impact of additional disconnections in the GB. They are most prominent in slightly asymmetric boundary segments observed in the experiment, compensating for deviations in GB plane inclination or a slight twist between both neighboring grains. One of the most prominent disconnections in the experimental datasets is highlighted in Supplementary Fig. 11. Its Burgers vector was determined as described for the phase junctions in the previous section to be $\mathbf{b}_{\text{exp.disc.}} = (?, 0.23, -0.06)$ Å and its full Burgers vector is thus close to the $(2.09, 0.21, -0.12)$ Å DSC vector as derived from the dichromatic pattern (see Supplementary Fig. 10). If disconnections compensate the GB asymmetry or a twist

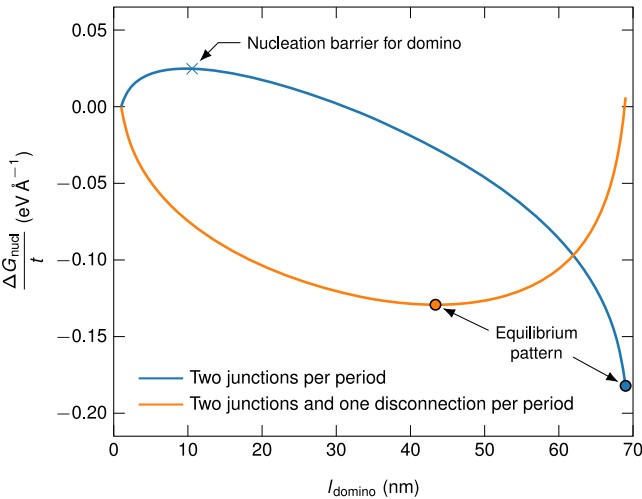

**Fig. 8 Free energy change due to a domino nucleus at $T = 400$ K.**
Calculated for a periodic system of alternating pearl and domino phases with a period $L = 70$ nm. Calculations for a system with only phase junction defects and for a system with an additional disconnection per period. We assume $\delta_0 = 1$ nm and subtract the value of $\Delta G/t$ at $l_{domino} = 1$ nm, since the offset of the curves depends on the exact value of the defect core energies.

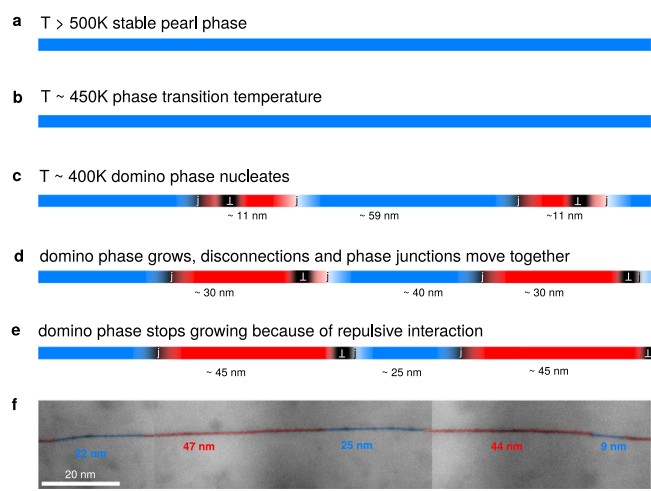

**Fig. 9 Schematic for the occurrence of the observed patterning of pearl and domino phase. a–e** The domino phase nucleates upon cooling from above the phase transition temperature. The newly formed phase junctions start interacting with pre-existing disconnections, leading to elastic repulsion and prevention of a complete transformation to the domino phase. **f** Nearly symmetric region from Fig. 2a shown for comparison.

between two grains, their Burgers vectors must be equal (if they were opposite, the average GB plane would remain symmetric or the twist would be undulating). The large $b_1$ component cannot be explained by a compensation of a GB asymmetry deviating from the symmetric {1 10 11} GB plane alone, but implies that a twist component along the ⟨1 1 1⟩ axis is compensated by such a disconnection. A disconnection occuring every 70 nm corresponds to a twist of about 0.2°. Such a small twist component is likely to occur between two neighboring grains due to the unavoidable, slight roughness of the substrate. The electron backscattered diffraction measurements showed deviations of up to 2° from the perfect ⟨1 1 1⟩ orientation of different grains (see Supplementary Fig. 1), indicating that twist components are likely to occur between two grains.

Figure 8 shows the free energy of a domino nucleus $\Delta G_{nucl}/t$ in a periodic pattern of alternating domino and pearl phases with a period of $L = 70 nm$ (fitting the experimentally observed period length and thus the average distance of the original nuclei). It was calculated with a finite amount of periodic images, but converges quickly with the number of images. The free energy is minimal when the whole pearl phase disappears since the GB phase junctions have attractive interactions. We introduced the experimentally observed disconnection, which is attracted by one of the junctions and repulsed by the other, by assuming it merges with one of the junctions into one combined Burgers vector. Here, a minimum appears at 40–50 nm segment length for domino and 20–30 nm for pearl, corresponding to the experimental patterning. An increase of the period length $L$ could be facilitated either by the disappearance of one or more pearl phase segments, which is connected to a high energy barrier in the patterned GB, or by elongation of the total GB length. In a real system, though, the GB length is typically fixed by triple junctions and would have to increase its curvature. The pattern is thus stabilized. It should also be noted that the relative regularity of the observed pattern supports our defect interaction hypothesis. If the pearl phase were left solely due to kinetic reasons, a more random arrangement would be expected.

The evolution of the patterning process is sketched in Fig. 9. At higher temperatures, the grain boundary consists of pearl phase

and possibly disconnections to compensate for slight twist components of the GB. During cooling, domino segments nucleate and disconnections are attracted to the junctions. The growth of the domino phase is stopped by the repulsive interaction between the combined junction/disconnection defect with the undecorated junctions and a GB phase pattern appears.

## Discussion

We observed two GB phases in a $\Sigma37c \langle 1\,1\,\bar{1} \rangle$ {1 10 11} GB in a 1-μm-thick elemental copper film by HAADF-STEM. Over a more than 300 nm long GB segment, these phases form an alternating pattern between segments of domino phase (40–100 nm long segments) and pearl phase (10–25 nm long segments). Free energy calculations on the structures simulated with an EAM potential show a diffusionless, congruent phase transition from domino (low temperature) to pearl (high temperature) at around 460 K. In light of this, the observation of patterning at room temperature is surprising, since the phase coexistence at ambient pressure is limited to a single temperature by Gibb's phase rule for elemental systems. Limited kinetics of the GB phase junction motion, but also the elastic interaction field of GB phase junctions and existing disconnections could play a significant role. Therefore, we quantitatively determined the Burgers vectors of the GB phase junctions and disconnections, which match the predictions from the differences in excess shears [**B**] and DSC lattice, respectively. By considering the elastic interactions between these defects, which resemble those of lattice dislocations, we found that certain arrangements of defects can energetically stabilize the phase pattern. While pure phase junctions occur in pairs with opposite Burgers vectors and thus have attractive interactions, which favor consolidation of a single phase in case of sufficient mobility, the addition of regularly spaced disconnections with the same Burgers vectors can support the patterning. Such disconnections can occur to compensate a slight twist component of the GB and are attracted to one half of the phase junction pair and repulse the other. It is known that GB phase transitions can influence material properties such as diffusion or GB mobility and we therefore expect that phase patterning could open up new ways to control such properties.

## Methods

**Specimen preparation.** The epitaxially grown copper thin films with a ⟨111⟩ surface orientation were deposited by molecular beam epitaxy (MBE) at the Central Scientific Facility Materials of the Max Planck Institute for Intelligent Systems in Stuttgart. To remove any contaminants from the substrate holder it was first annealed at 1273 K for 1 h without substrate. Subsequently, a single crystalline ⟨0001⟩ $Al_2O_3$ wafer with a miscut of <0.1° from CrysTec was inserted into the ultra-high vacuum (UHV) chamber and sputter cleaned for 5 min with a 200 eV $Ar^+$ ion beam to remove residues on the surface. During the sputtering process, the wafer was rotated at 20 rpm. The wafer was then directly annealed in the vacuum chamber at 1273 K for 1 h to reconstruct the substrate surface and remove any residual contamination. Then, the copper thin film with a final thickness of 1 μm was deposited by MBE at room temperature with a deposition rate of 0.05 nm s$^{-1}$ at a chamber pressure of $\sim 5 \cdot 10^{-10}$ mbar to a final thickness of 1 μm. In a final step, the deposited film was annealed at 673 K for 1 h under high vacuum conditions without venting the chamber.

Characterisation of the microstructure has been performed using a Thermo Fisher Scientific Scios2HiVac dual-beam SEM equipped with an EBSD detector. Two inverse pole figure maps of electron backscatter diffraction scans are shown in Supplementary Fig. 1. Site-specific plane-view FIB lamellas have been lifted out and thinned using the Scios 2 DualBeam SEM/FIB microscope, starting with a gallium ion beam voltage and current of 30 kV and 0.1 nA and ending at 5 kV, 16 pA.

**Scanning transmission electron microscope.** The FIB lamellas were investigated with a probe-corrected FEI Titan Themis 80-300 (Thermo Fisher Scientific). The electrons, which are emitted by a high-brightness field emission gun, were accelerated to 300 kV. The probe current has been set to 70–80 pA. The STEM datasets were registered with a high-angle annular dark-field (HAADF) detector (Fishione Instruments Model 3000), using collection angles of 78–200 mrad and a semi-convergence angle of 17mrad. The datasets consist of image series with 50–100 images and a dwell time of 1 μs. In order to reveal the structures, as well as to reduce noise and instabilities of the instruments, these datasets have been averaged and optimized by using a background substraction filter, Butterworth filter and Gaussian filter, ensuring that the atomic structure of the original image has been preserved.

**MD simulations.** MD simulations were performed using LAMMPS[46] (https://lammps.sandia.gov/) with an EAM potential for Cu by Mishin et al.[34]. This potential has proven to at least qualitatively capture the structures of ⟨111⟩ tilt grain boundaries before[28] and was developed with good agreement to the phononic properties of copper[34], which is important for the free energy calculations. All molecular dynamics simulation were performed with a time integration step of 2 fs.

For a simple structure search using the γ-surface method, we constructed a bicrystal out of two fcc crystallites with a size of 31.1 × 45.0 × 6.26 Å³ each (1312 atoms in total). The bottom crystal was oriented with [3̄ 4 7], [11 1̄ 10], and [1 1 1̄] along the x, y, and z directions. The top crystal orientation was [3 4 7], [11 1̄0 1], and [1 1 1̄], resulting in a misorientation angle of 50.57°. The x and z directions were periodic and their length was kept fixed to preserve the ground state fcc lattice constant of a = 3.615 Å, while the y direction contained open boundaries. We systematically displaced the top crystal and minimized the energy of the system to sample those GB configurations which do not require additional interstitial or vacancy atoms.

The lattice constant of fcc copper as a function of temperature was obtained by equilibration for 250 ps at each temperature in the isothermal-isobaric ensemble using a thermostat and a barostat at zero pressure[42]. Annealing simulations were performed with the same boundary conditions as the 0 K simulations above, but with the lattice constant adjusted to the target temperature and with a larger box size of 630 × 160 × 63.5 Å³ (526,800 atoms). One simulation with open boundaries in x direction was performed to investigate possible phase transitions requiring a particle reservoir[15]. This simulation cell was chosen to be thicker in y direction and shorter in x direction (370 × 230 × 63.5 Å³, 435,320 atoms), to avoid any influence due to the diffusion-driven changes on the surface during the long annealing time of 30 ns.

All simulation results were visualized with OVITO[47].

**Structure search and calculation of excess properties.** Possible GB structures were sampled with an evolutionary algorithm[25] implemented using the USPEX code[37,38].

For these samples, excess properties were calculated in a region around the grain boundary excluding atoms closer than 1 nm to the surface. We use the definition of the excess properties by Frolov and Mishin[36,43]. The excess number of atoms [n] is defined in terms of a fraction of a {1 10 11} plane[15] as

$$[n] = \frac{N}{N_{\{1\,10\,11\}}} \bmod 1 \qquad (8)$$

where N is the total number of atoms in the simulation cell and $N_{\{11011\}}$ corresponds to the number of atoms in a defect-free {1 10 11} plane of the fcc crystal.

The microscopic translation vector [B] between the two crystallites was computed by first constructing a dichromatic pattern in the first crystallite far away from the grain boundary and extending this pattern to the second crystallite. Now, [B₁] and [B₂] were obtained by shifting the dichromatic pattern with fixed [B₃] = [V] to fit the second crystallite. In case of [n] = 0 this is sufficient, but interstitial-like atoms or vacancy-type defects in the grain boundary can also affect [B₃] (Supplementary Fig. 3c). For these cases, we also varied [B₃] to obtain a fit. The values of [B] were then restricted to the DSC unit cell.

In order to separate the pearl and domino phases, we used a k-means clustering algorithm as implemented in scikit-learn[48] on the γ₀, [V], [τ₁₁], [τ₂₂], |[τ₁₂]|, [n], and [B₁] data. The [B₂] data does not exhibit any pattern and was excluded. The silhouette coefficient[49] indicates optimal clustering at k = 2 clusters (Supplementary Fig. 3d).

**STEM image simulation.** STEM image simulations were performed using the multislice algorithm implemented in the package abTEM[50,51]. The atomistic simulation cells were imported with the Python library ASE[52] and oriented in such a way that the [11 1̄] direction coincides with the electron direction in the STEM simulations. An electron probe with 300 keV, a semi-angle of 17.8 mrad, a focal spread of 100 and a defocus of 0 was given to match the settings used in the experiment. The HAADF detector was set to 77.9–200 mrad. The step size was selected as 0.178 Å to match to the imaging conditions. A slice thickness of 2 Å was used as the atomic column separation in z direction (the tilt axis) is 2.09 Å. All simulated cells were of the same thickness of 63 Å to ensure comparability.

**Free energy calculation.** Free energies were calculated using the quasi-harmonic approximation[40,41]. Phononic eigenfrequencies were obtained from force constant matrices computed with the `dynamical_matrix` command in LAMMPS. For each GB, a corresponding fcc slab was produced with the same number of atoms and the same surfaces. This is necessary, since the subsystem method described by Freitas et al.[41] introduces an artificial boundary in the force constant calculation. The GB free energy γ(T) is then simply the free energy difference between these two systems normalized to the grain boundary area. We confirmed the results using thermodynamic integration[42] along the Frenkel–Ladd path[53] with the subsystem method[41]. These results agree well except for the pearl #1 structure (Supplementary Fig. 6). The pearl #1 structure started nucleating the B motif of the pearl #2 structure at temperatures above 200 K, leading to large dissipation during the thermodynamic integration path. The data was discarded due to its unreliability. This behavior of the pearl phase is nevertheless in accordance with the expected phase stability predicted using the quasi-harmonic approximation: The pure pearl #1 structure is less stable than pearl #2 (Fig. 5).

**Burgers circuit.** A Burgers circuit is drawn around a phase junction as described by Frolov et al.[31]: The Burgers circuit is split into 4 vectors—2 vertical vectors, crossing the domino and the pearl phase each, and 2 horizontal vectors, described by specific planes in each grain next to the GB. The starting and end points of the vectors across the GB phases are related to recognizable features in the GB structures (green markers in Fig. 7c–e). The same vectors are measured in regions far away from a phase junction, to aim for a stress-free reference structure. Therefore, images taken in an area without phase junction are used and the atomic positions of the same recognizable features are localized by applying a Gaussian peak fitting algorithm[54]. They are averaged over at least four identical sites in the reference states. Thereby, differences up to 0.5 Å are observed, which could be limited to an uncertainty of ±0.1 Å since we averaged over several measurements.

In the original description of the method[31], the horizontal lines in both grains are parallel to the GB plane and cancel each other out. Here, this is not the case as the planes needed to be shorter and still well defined. Since these lines are completely in defect-free regions, we can determine the corresponding vector in the crystal coordinate system by counting atomic columns along specific crystallographic directions. For the example shown in Fig. 7d, these are $c_\mu = a[-34/6 \ -46/6 \ -80/6]$ in the upper and $c_\lambda = a[-34/6 \ 80/6 \ 46/6]$ in the lower grain. To add them up, we rotate the line in the lower grain into the crystal coordinate system of the upper grain[55,56] to obtain

$$\mathbf{b}_\mu = \mathbf{c}_\mu + M_1 \mathbf{c}_\lambda, \qquad (9)$$

where $M_1$ is the corresponding rotation matrix. In a second step, $\mathbf{b}_\mu$ needs to be separated into components normal and parallel to the GB, so it is again multiplied by an appropriate rotation matrix $M_2$. An error of ±0.1 Å was identified for these values taking into account the limitation of measuring the misorientation between both grains and the GB plane (±0.3°). The Burgers vector results as the sum of all 4 vectors—the parts crossing each phase in a reference state as well as the horizontal parts in both grains.

## Data availability

The main datasets generated in this study have been deposited in the Zenodo database under accession code https://doi.org/10.5281/zenodo.5354071[57].

## Code availability

Custom code to perform MD simulations, analyse simulation structures, and compute disconnection interactions is available together with the published datasets

at https://doi.org/10.5281/zenodo.5354071[57]. Publicly available software packages that were used in this work are listed in the relevant Methods sections. All other code is available upon request.

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

## Acknowledgements

The authors thank Niels Cautaerts for helpful discussions and help with the STEM image simulation. G. Richter and his team from the Max Planck Institute for Intelligent Systems are gratefully acknowledged for producing the Cu thin film by molecular beam epitaxy. This project has received funding from the European Research Council (ERC) under the European Union's Horizon 2020 research and innovation programme (Grant agreement No. 787446; GB-CORRELATE). This work was partially performed under the auspices of

the U.S. Department of Energy (DOE) by the Lawrence Livermore National Laboratory (LLNL) under Contract No. DE-AC52-07NA27344. T.F. acknowledges the funding by the Laboratory Directed Research and Development Program at LLNL under Project Tracking Code number 19-ERD-026.

## Author contributions

T.B. and L.L. contributed equally to this work. L.L. performed the experimental sample preparation, HAADF-STEM investigations and analysis of the obtained datasets. C.H.L. and G.D. designed the concept of the experimental study. T.B. and R.F. calculated free energies using the quasi-harmonic approximation and T.F. conducted the USPEX simulations. All other simulations and analyses of the simulation data were performed by T.B. The project was supervised by C.H.L. and G.D., who also contributed to discussions. G.D. secured funding for L.L. and T.B. via the ERC grant GB-CORRELATE. L.L. and T.B. prepared the initial draft and all authors contributed to the preparation of the final manuscript.

## Funding

## Competing interests

The authors declare no competing interests.
