## [Peer Review File · Nature Communications]

Title: Dual phase patterning during a congruent grain boundary phase transition in elemental copperREVIEWER COMMENTS

Reviewer #1 (Remarks to the Author):

The present paper is a follow-up from Meiners et al. (2020). The authors present a detailed explanation, involving both calculations and the analysis of experimental data, for the co-existence of two grain boundary complexions at room temperature, even though one of them has a free energy advantage. The quality of the work is exceptional. The patterned structure is kinetically stabilized by disconnections that interact differently with the adjacent interface segments that separate the two boundary phases. Taken together, this explanation is consistent with the results presented. My only comment is that a very two dimensional picture is presented. Are we to imagine that the pearl and domino phases exist in stripes across the boundary? Or is it more like a maze bicrystal? If it is the latter, it is hard to imagine how a semi-regular pattern would be observed in two dimensions.

Reviewer #2 (Remarks to the Author):

The submission titled “Dual phase patterning during a congruent grain boundary phase transition in elemental copper” discusses an experimental and computational study on grain boundary ‘phase’ or ‘complexion’ transformations along/through a grain boundary, particularly regarding stability, nucleation, and growth. Among the key results was that two different atomistic structures located within a $\Sigma 37c \langle 111 \rangle \{1011\}$ grain boundary coexist at several temperatures, and that there is a reversible transformation upon heating or cooling. In addition, the authors proposed and validated a hypothesis that GB phase lengths are stabilized by defect interactions within the grain boundaries. Another interesting observation was that phase junctions occurred in the center of the grain boundary, meaning that a grain boundary ‘phase’ transformation does not necessarily nucleate primarily on triple junctions as one may conventionally expect. Overall, the manuscript was an enjoyable read, the data was concrete and exemplary, the Figures were very nicely prepared, and the submission as a whole is worthy of publication in Nature Communications. The work is also of significance because it discusses a grain boundary ‘phase’ or ‘complexion’ transformation through a grain boundary, in which very little is known, and how such transformations propagate through boundaries, and eventually, through adjacent triple junctions.

Listed below are comments that should be considered:

1. There is insufficient information about the film growth. More specifics should be added to the methods section that can strengthen the reproducibility of the work.
2. One specific question is that while the films are intended to be pure Cu, there is a strong possibility that impurity species were inserted into the film and eventually segregated to the grain boundaries? If so, the Gibbs phase rule changes since there is an additional degree(s) of freedom. Perhaps this could be an alternative reason for the unexpected/surprise observation of GB phase patterning at room temperature... I also suspect that impurities would segregate to the phase junctions, thereby also

leading to slower mobilities/kinetics via a drag effect. In general Supplementary EDS, EELS, or APT data would greatly strengthen the work by re-affirming that the grain boundaries are practically 'pure'.

3. Based on the supplemental Figure S1, there is a moderate fraction of Σ_{19b} grain boundaries. Were these interfaces investigated? If so, did they exhibit similar domino or pearl phases? Otherwise, how many separate $\Sigma_{37c} \langle 111 \rangle \{1011\}$ grain boundaries were imaged and did their structures match that of the grain boundary discussed in this work?

4. I enjoyed the quantitative description of the GB segment discussion in the second paragraph of IIA, as well as the clustering analysis described in Figure 4.

5. The lowest-energy pearl phase was not experimentally observed. Any specific reason why? The domino phase also appears to be more ordered, so could the pearl structure be more entropically stabilized at higher temperatures? This may also indirectly explain why the pearl phase is more prone to containing defects.

6. The STEM-HAADF simulations support the work very nicely. However, I imagine that it was not trivial to simulate the domino and pearl structures... How exactly were structures from the UPSEX simulations prepared to run the simulations? Were they converted to .xyz files or something similar? More detail would be helpful.

7. Is there any evidence to suggest that the different domino or pearl grain boundary 'phases' have different characteristic properties that significantly affect bulk materials (e.g. strength, conductivity, etc.)? Otherwise, the authors state in the conclusions that GB phase patterning and GB phase transitions may influence properties such as diffusion or GB mobility, but it is unclear if similar domino and pearl GB phases are expected to be observed in more industrially significant multi-component alloys? Are the domino and pearl GB phases more expected to exist in elemental Cu (and perhaps other FCC metals)?

Reviewer #3 (Remarks to the Author):

The authors have conducted a thorough study of the phase transition of a Σ_{37} GB by experimental analysis and atomistic simulations. This work is very helpful to further understand the diversity of grain boundary structure, especially for understanding the stable and metastable structures of the grain boundary with the same misorientation angle. The manuscript is well organized, and the research methods are state of the art and are technical no problems. Two issues that require further consideration and explanation before it is publication.

(1) In the study of grain boundary structure, the authors use the same experimental and simulation methods in the current study and the previous studies on other GBs. For example, the same evolutionary algorithm and clustering method of this study are used in the previous study (Ref. [25]), the similar GB phase structure ('domino' and 'pearl') are observed in $\Sigma_{37c} \langle 111 \rangle \{1 10 11\}$ GB of this work

and in $\Sigma 19b\langle 111 \rangle\{1\bar{7}8\}$ GB of the published work (Ref. [28]). It seems that the author used the same method to study a different object, and why chose this particular grain boundary as the research object is not well addressed.

(2) In section C, the authors have the conclusion that the GB phase transition is a diffusionless process, which seems inadequate. Since there are no atoms added or deleted in the pristine GB, it is hard to understand how the GB structure transformation occurred without the temperature effect and the activation of the thermal diffusion mechanism. As the authors discussed in section E, I think the temperature gradient plays an important role in the GB phase transition. It is recommended to illustrate the mechanisms of the GB phase transition by MD simulation.

Answers to the remarks of the reviewers

Reviewer #1:

“ The present paper is a follow-up from Meiners et al. (2020). The authors present a detailed explanation, involving both calculations and the analysis of experimental data, for the co-existence of two grain boundary complexions at room temperature, even though one of them has a free energy advantage. The quality of the work is exceptional. The patterned structure is kinetically stabilized by disconnections that interact differently with the adjacent interface segments that separate the two boundary phases. Taken together, this explanation is consistent with the results presented. My only comment is that a very two dimensional picture is presented. Are we to imagine that the pearl and domino phases exist in stripes across the boundary? Or is it more like a maze bicrystal? If it is the latter, it is hard to imagine how a semi-regular pattern would be observed in two dimensions. ”

We thank the reviewer for their positive recommendation.

It is true that a fully three-dimensional view into the arrangement of the grain boundary phases would be desirable. However, since we are experimentally limited to a two dimensional view in a scanning transmission electron microscope, it is difficult to access the full 3D atomic scale information of the grain boundary across the entire thin film. In order to learn more about the structure of the phases along the tilt axis, we lifted out an area parallel to the $\langle 111 \rangle$ axis of the $\Sigma 37c$ grain boundary (GB) over the complete film thickness. We could observe the lattice plane shift of the domino phase solely and we did not find any indication towards a GB phase junction. Thus, one GB phase should be present along the entire thickness of the film at that position and no maze-like structure could be observed.

One possible explanation for an anisotropic growth of the GB phases may be related to the anisotropy in the core energy of the GB phase junctions, which is mentioned in our paper (pg. 8, l. 395ff): Winter et al. [Nucleation of grain boundary phases, PRL 128 (2022)] calculated the direction-dependent core energy of grain boundary phase junctions in a tungsten $\Sigma 29$ (520) [001] GB. It is shown that the core energy is four times lower in the tilt direction than in its normal direction in the GB plane. Furthermore, Frolov and Mishin [Phases, phase equilibria, and phase rules in low-dimensional systems, J. Chem. Phys. 143 (2015)] showed a GB phase transformation in a $\Sigma 5$ {100} (210) GB in copper and the GB phase nucleus was observed to be anisotropic, showing an elongated shape in the tilt direction. Thus, it is likely that also in our case the GB phases appear in stripes and not in a maze like structure. The discussion about the anisotropy of the nucleation of secondary phases can be found in the results subsection “Mechanisms leading to grain boundary phase patterning” (pg. 8, l. 395ff) and we added the following note:

pg. 8, l. 411ff: “We note here that the present investigation is limited to a rather two-dimensional picture and it remains currently unclear whether such a striped GB phase pattern would change, for example, into maze-like patterns in a large-grained bulk polycrystal. In order to observe the 3D arrangement of GB phases within the GB plane, new imaging modalities in the TEM would need to be developed.”

Reviewer #2:

“ The submission titled “Dual phase patterning during a congruent grain boundary phase transition in elemental copper” discusses an experimental and computational study on grain boundary ‘phase’ or ‘complexion’ transformations along/through a grain boundary, particularly regarding stability, nucleation, and growth. Among the key results was that two different atomistic structures located within a $\Sigma 37c \langle 111 \rangle \{1\ 10\ 11\}$ grain boundary coexist at several temperatures, and that there is a reversible transformation upon heating or cooling. In addition, the authors proposed and validated a hypothesis that GB phase lengths are stabilized by defect interactions within the grain boundaries. Another interesting observation was that phase junctions occurred in the center of the grain boundary, meaning that a grain boundary ‘phase’ transformation does not necessarily nucleate primarily on triple junctions as one may conventionally expect. Overall, the manuscript was an enjoyable read, the data was concrete and exemplary, the Figures were very nicely prepared, and the submission as a whole is worthy of publication in Nature Communications. The work is also of significance because it discusses a grain boundary ‘phase’ or ‘complexion’ transformation through a grain boundary, in which very little is known, and how such transformations propagate through boundaries, and eventually, through adjacent triple junctions.

Listed below are comments that should be considered:

1. There is insufficient information about the film growth. More specifics should be added to the methods section that can strengthen the reproducibility of the work.

We appreciate the referee’s comments and following their suggestion we added more specific information regarding film growth in the Methods section of the manuscript on pg. 10, l. 577ff:

“The epitaxially grown copper thin films with a $\langle 111 \rangle$ surface orientation were deposited by molecular beam epitaxy (MBE) at the Central Scientific Facility Materials of the Max Planck Institute for Intelligent Systems in Stuttgart. To remove any contaminants from the substrate holder it was first annealed at 1273 K for 1 h without substrate. Subsequently, a single crystalline $\langle 0001 \rangle$ Al_2O_3 wafer with a miscut of less than 0.1° from CrysTec was inserted into the ultra high vacuum (UHV) chamber and sputter cleaned for 5 min with a 200 eV Ar^+ ion beam to remove residues on the surface. During the sputtering process, the wafer was rotated at 20 rpm. The wafer was then directly annealed in the vacuum chamber at 1273 K for 1 h to reconstruct the substrate surface and remove any residual contamination. Then, the copper thin film with a final thickness of $1\ \mu\text{m}$ was deposited by MBE at room temperature with a deposition rate of $0.05\ \text{nm/s}$ at a chamber pressure of $\sim 5 \cdot 10^{-10}$ mbar to a final thickness of $1\ \mu\text{m}$. In a final step, the deposited film was annealed at 673 K for 1 h under high vacuum conditions without venting the chamber.”

“ 2. One specific question is that while the films are intended to be pure Cu, there is a strong possibility that impurity species were inserted into the film and eventually segregated to the grain boundaries? If so, the Gibbs phase rule changes since there is an additional degree(s) of freedom. Perhaps this could be

an alternative reason for the unexpected/surprise observation of GB phase patterning at room temperature. . . I also suspect that impurities would segregate to the phase junctions, thereby also leading to slower mobilities/kinetics via a drag effect. In general Supplementary EDS, EELS, or APT data would greatly strengthen the work by re-affirming that the grain boundaries are practically ‘pure’.

We thank the referee for this important comment. Since we have used high purity copper and deposited as well as annealed the film under ultra high vacuum conditions it is unlikely that major impurities or segregation was introduced during sample fabrication. Furthermore, our experimental observations match nearly perfectly with atomistic simulations for pure Cu, suggesting that solutes do not impact the observed grain boundary phase patterning.

However, as suggested by the referee, we conducted additional high resolution EDS and EELS experiments. High resolution EDS experiments combined with HAADF-STEM imaging allows relating the spectra to the GB and phase junction area as shown in Fig. R1 below. The HAADF-STEM image and corresponding rectangular regions from where the EDS spectra are extracted are shown in Fig. R1a. The intensities of integrated spectra at the phase junction as well as along the complete GB are shown in Fig. R1b. The intensity is normalized each time to the maximum intensity values. Two major peaks are visible which can be attributed to Cu. In Fig. R1c we plot the difference spectra obtained by subtracting the integrated spectrum of a region around the GB phase junction (orange box in Fig. R1a) from a region with the same dimensions within the grain (pink box in Fig. R1a). We did the same for the integrated spectra along the complete GB, subtracting the integrated spectrum of the GB (turquoise box) from a reference area (purple box). Both difference spectra are close to zero. Significant changes are visible only at the slopes of the major Cu peaks. Thus we see that both EDS spectra, at the phase junction, but also at the complete GB do not differ from regions within the grain and we can exclude segregation of any element at the GB or the phase junction. Nevertheless, we wanted to identify all detected peaks in a sum spectrum of the complete area in a next step. As shown in Fig. R1d, major Cu $L\alpha$, $K\alpha$ and $K\beta$ peaks are visible in the EDS spectrum. A smaller oxygen peak can be attributed to Cu oxides at the sample surface that formed during sample transport. The carbon peak at about 0.277 keV most likely originates from hydrocarbon contamination on the sample, which is broken down to carbon during electron beam irradiation in the STEM. One minor peak of silicon is related to our X-rays generated in the Si-drift EDS detector (Fig. R1d). If we scale the X-ray spectrum to magnify low intensity features shown in Fig. R1e, we find minor peaks of Fe and Zr, which are stemming from stray X-rays in the pole piece of the microscope. The additional peaks at 2.4 keV (CB-2), 3.6 keV (CB-3) and even 4.8 keV (CB-4) are related to coherent Bremsstrahlung peaks from the (112) planes in Cu, which may occur when the sample is oriented in zone axis orientation.

To be more sensitive for potentially low impurity concentrations and light elements such as C and O, we performed STEM-EELS measurements across the GB shown in Fig. R2. Also here, we found no peaks besides oxygen and carbon, which can be most likely attributed to surface oxides as well as small contamination formed during sample preparation. A hint of the Cu-M2 peak is visible at 74 eV. Again, we compared an area in the grain boundary with an area of same dimensions within the grain analogously to the EDX investigations. As visible in the low loss spectrum in Fig. R2b and the core loss spectrum in Fig. R2c, the signal in the grain and at the GB are similar. Thus, we can again exclude segregation of any element at the GB.

We included the following in the manuscript: pg. 10 l. 626ff: “STEM-EDS and STEM-EELS investigations were performed (not shown here) in order to exclude the segregation of solutes to the GB or phase junctions. Within the detection limits of the respective techniques, no segregation to the GB or GB phase junction could be observed.”

Figure R1: EDS analysis of a $\Sigma 37c$ GB. (a) Corresponding HAADF-STEM image during EDS analysis. (b) Sum spectra of selected areas as shown in (a). (c) Subtraction of the intensity of the phase junction / GB with a reference area of the same size inside a grain. (d)–(e) Sum spectrum of the complete area present in (a) with labelled peaks.

Figure R2: EELS analysis of a $\Sigma 37c$ GB. a) ADF image of the area investigated by EELS. The bright vertical line is the GB. b) and c) Low loss and high loss spectra of the areas as indicated in a). To be able to see both signals separately, the counts of the GB area were multiplied by a factor of 1.5, which shifted the whole signal in y-direction.

“ 3. Based on the supplemental Figure S1, there is a moderate fraction of $\Sigma 19b$ grain boundaries. Were these interfaces investigated? If so, did they exhibit similar domino or pearl phases? Otherwise, how many separate $\Sigma 37c$ $\langle 111 \rangle$ $\{1\ 10\ 11\}$ grain boundaries were imaged and did their structures match that of the grain boundary discussed in this work? ”

The $\Sigma 19b$ grain boundaries were already investigated in a film with similar deposition parameters in a previous study by Meiners et al. [Observations of grain-boundary phase transformations in an elemental metal, *Nature* 579, 375 (2020)]. In that study, the $\Sigma 19b$ GBs did indeed exhibit two GB phases observed at symmetric and asymmetric GB segments, which are similar to the ones observed in this paper. However, there are several significant and interesting differences: according to the modelling results, the GB phase transition could only be established by applying stresses to the GB in the $\Sigma 19b$ GBs, since the domino phase is metastable with respect to the pearl phase. In the $\Sigma 37c$ case, we showed that temperature alone is sufficient in the model to transform pearl into domino and vice versa. This is also a potential explanation why multiple pearl and domino segments are observed here, offering the potential to study the evolution of grain boundary phases experimentally.

For the analysis of the $\Sigma 37c$ GB, we could reveal the atomic structure of three different grain boundaries at different locations in our Cu thin film as shown in the Supplemental Fig. S1, where two out of the three investigated GBs are indicated. The atomic structures of both phases were the same in all different GBs, showing each time an alternation of domino and pearl phase. Furthermore, the overview picture of 200 nm length shows that the observation of these two phases does not occur only once but multiple times. All the times, the structures are similar and match nicely to the predicted structures by simulation.

We added the following part in the manuscript:

p. 2, l. 83ff: “Plane-view focused ion beam (FIB) lamellas of $\Sigma 37c$ $\langle 111 \rangle$ GBs were lifted out from two different positions in the film and their atomic structures were investigated by high-angle annular dark-field (HAADF) STEM.”

“ 4. I enjoyed the quantitative description of the GB segment discussion in the second paragraph of IIA, as well as the clustering analysis described in Figure 4. ”

We thank the referee for acknowledging our experimental as well as modelling efforts.

“ 5. The lowest-energy pearl phase was not experimentally observed. Any specific reason why? The domino phase also appears to be more ordered, so could the pearl structure be more entropically stabilized at higher temperatures? This may also indirectly explain why the pearl phase is more prone to containing defects. ”

The lowest-energy pearl structure (#1) has a lower free energy only up to around 450 K, which is very close to the transition from the low-temperature domino phase to the high-temperature pearl phase (see Fig. 5 in the main text). Above 450 K, the structure observed in the experiment (pearl #2) is predicted to have a lower free energy than both domino and pearl #1 and is thus more stable. Thus, pearl #1 is never the stable phase at any temperature. Furthermore, as shown by a HAADF-STEM simulation (Supplemental Fig. S6), even if a fraction of pearl #1 phase would occur, a mixture of pearl #1 and #2 motifs cannot be differentiated from a pure pearl #2 phase in the experiment. Thus, it is possible that a mixed pearl phase occurs but could just not be resolved, which is why we opted to regard the pearl variants as microstates of the pearl phase and not as two different phases.

Regarding the entropy of the domino and pearl phases: We thank the referee for this very important comment. We can directly estimate the excess grain boundary entropy via $[S] \approx -d\gamma/dT$ [see also Freitas et al., Free energy of grain boundary phases, PR Mater. 2 (2018)]. We get values of approximately 0.21 mJ/(m²K) for domino and 0.24 mJ/(m²K) for pearl #1 and #2 at $T = 0$ K. This shows exactly what the reviewer pointed out – the pearl phase has a higher vibrational entropy, which is responsible for the phase transition. Since this is an interesting point that was not explicitly mentioned, we expanded the discussion of the phases’ relative entropy in the updated text:

pg. 5 l. 226ff: “This indicates that the pearl phase is a structure with higher entropy. We estimated the excess entropy of the GB phases via [36,41,43] $[S] \approx -d\gamma(T)/dT$. At $T = 0$ K, we obtain excess entropies of approximately 0.21 mJ/(m²K) for domino and 0.24 mJ/(m²K) for both pearl structures, confirming the higher entropy of pearl.”

“ 6. The STEM-HAADF simulations support the work very nicely. However, I imagine that it was not trivial to simulate the domino and pearl structures...How exactly were structures from the USPEX simulations prepared to run the simulations? Were they converted to .xyz files or something similar? More detail would be helpful. ”

We fully agree with the referee that STEM image simulations provide an ideal link between experiment and modelling, but also that the entire workflow should be presented in detail for reproducibility. Therefore, we added more specific information in the Methods section. In principle, we used the simulation cells retrieved from the USPEX analysis, which are datasets with xyz-coordinates for each atomic position. We ensured the correct orientation of the simulation cell by rotating the cell in such a way that the z -direction corresponds to the $\langle 111 \rangle$ tilt axis using OVITO and saved the dataset in POSCAR file format. For reading the files, the Python library ASE was used, which can read multiple common formats (note that it is therefore not too important which file format was used to store the data). In a second step, with the help of the Python library abTEM, we performed the HAADF-STEM image simulation. The parameters needed for running a multi-slice HAADF-STEM simulation are described in the Methods section. The updated text reads as follows:

pg. 11, l. 709ff: “STEM image simulations were performed using the multislice algorithm implemented in the package abTEM [50,51]. The atomistic simulation cells were imported with the Python library ASE [52] and oriented in such a way that the $[11\bar{1}]$ tilt axis coincides with the electron beam direction in the STEM simulations. An electron probe with 300 keV, a semi-angle of 17.8 mrad, a focal spread of 100 Å and a defocus of 0 was given to match the settings used in the experiment. The HAADF detector was set to 77.9–200 mrad. The step size was selected as 0.178 Å to match to the imaging conditions. A slice thickness of 2 Å was used as the atomic column separation in z direction (the tilt axis) is 2.09 Å. All simulated cells were of the same thickness of 63 Å to ensure comparability.”

“ 7. Is there any evidence to suggest that the different domino or pearl grain boundary ‘phases’ have different characteristic properties that significantly affect bulk materials (e.g. strength, conductivity, etc.)? Otherwise, the authors state in the conclusions that GB phase patterning and GB phase transitions may influence properties such as diffusion or GB mobility, but it is unclear if similar domino and pearl GB phases are expected to be observed in more industrially significant multi-component alloys? Are the domino and pearl GB phases more expected to exist in elemental Cu (and perhaps other FCC metals)? ”

Indeed these are very interesting and relevant questions. However, they require careful and concurrent observation of both GB structure and macroscopic properties, which is the subject of ongoing studies in our group. Here, see e.g. Bishara et al. [Understanding Grain Boundary Electrical Resistivity in Cu, ACS Nano 15 (2021)] for some first results that suggest a connection between GB excess volume and resistivity, although GB structures could not be resolved experimentally.

In general, the phenomenon of grain boundary (GB) phase transformations and observation of different GB phases is very new and only few studies exist observing two different GB phases within the same GB. Thus, the impact of such transitions on materials properties is still barely understood. Some time ago, Divinski et al. could already observe the anomalous diffusion of Ag and Au in $\Sigma 5$ [001] tilt GBs in copper [Divinski et al., Diffusion and segregation of silver

in copper $\Sigma 5(310)$ grain boundary, PRB 85, 144104 (2012)]. Atomistic modelling performed by Frolov et al. [Effect of Interface Phase Transformations on Diffusion and Segregation in High-Angle Grain Boundaries, PRL 110, 255502 (2013) and Structural phase transformations in metallic grain boundaries, NC 1899 (2013)] indicated that a first order grain boundary phase transformation can be responsible for the observed differences in diffusion, since the change in excess properties of the GB phases influences both the Ag segregation behavior and diffusion.

Since in our study both GB phases have different excess volume, this could also have a strong impact on the diffusion or segregation behaviour of different elements. Furthermore, due to the patterning of two phases, a dense network of GB phase junctions is observed. This might largely influence the dislocation slip behaviour through the grain boundary. The introduction of alloying elements is ongoing research and out of scope for the present paper. We believe, though, that it is important to first understand pure elements and their behavior before going to a more complex system like a multi-component alloy. If similar domino and pearl GB phases are observed in other pure and alloyed fcc metals is as well a subject of ongoing work.

Reviewer #3:

“ The authors have conducted a thorough study of the phase transition of a $\Sigma 37$ GB by experimental analysis and atomistic simulations. This work is very helpful to further understand the diversity of grain boundary structure, especially for understanding the stable and metastable structures of the grain boundary with the same misorientation angle. The manuscript is well organized, and the research methods are state of the art and are technical no problems. Two issues that require further consideration and explanation before it is publication. ”

(1) In the study of grain boundary structure, the authors use the same experimental and simulation methods in the current study and the previous studies on other GBs. For example, the same evolutionary algorithm and clustering method of this study are used in the previous study (Ref. [25]), the similar GB phase structure ('domino' and 'pearl') are observed in $\Sigma 37c\langle 111 \rangle \{1\ 10\ 11\}$ GB of this work and in $\Sigma 19b\langle 111 \rangle \{1\ 7\ 8\}$ GB of the published work (Ref. [28]). It seems that the author used the same method to study a different object, and why chose this particular grain boundary as the research object is not well addressed.

We thank the reviewer for this valuable feedback. It is correct that similar methods were used in parts of both studies, even though we would like to emphasize that determining the Burgers vector of the phase junction as well as explaining the patterning mechanism are very new in this study and were not investigated on the $\Sigma 19b$ grain boundaries. Moreover, the experimental observation of a grain boundary that is patterned by its grain boundary phases in a simple elemental metal has never been reported before (neither in simulations). The employed experimental and simulation methodologies are state-of-the-art and especially the evolutionary algorithm used here is the only one to our knowledge that efficiently samples the vast microstates of possible grain boundary phases.

Furthermore, we discovered new phenomena that appeared sufficiently different from previous work: In $\Sigma 19b$, a thermally-induced phase transformation has not been observed. The co-existence

of both phases could be only realised in simulations when stress or strain was applied to the system. The GB energy of the pearl phase was lower over the complete temperature range and the domino phase was only observed in short segments in the experiment. Here, we get a very different picture. Simulations predict that the phase transformation can be induced thermally and we experimentally observe a stable and regular patterning of both phases over a 200 nm segment. We could observe a nucleation of the pearl phase within the domino phase and indicate the Burgers vector for the phase junction. Evaluating disconnections which are present in our experimental observations, we calculated how their interaction fields could influence the phase junctions during a GB phase transformation process. All these results are very new compared to the previous study.

We highlighted this in the updated introduction:

pg. 1, l. 66ff: “The misorientation between both grains changed by only 4° compared the earlier study of GB phases in a $\Sigma 19b$ GB [28], but while similar GB phases occur, their thermodynamic stability is markedly different: Here, one GB phase is stable in low temperature regimes (below 460 K, called "domino phase") whereas the other GB phase (called "pearl phase") is energetically favored at elevated temperatures. The GB phase transformation can thus occur under ambient pressure by temperature alone.”

“ (2) In section C, the authors have the conclusion that the GB phase transition is a diffusionless process, which seems inadequate. Since there are no atoms added or deleted in the pristine GB, it is hard to understand how the GB structure transformation occurred without the temperature effect and the activation of the thermal diffusion mechanism. As the authors discussed in section E, I think the temperature gradient plays an important role in the GB phase transition. It is recommended to illustrate the mechanisms of the GB phase transition by MD simulation. ”

We are more than happy to clarify the GB phase transformation as well as the diffusionless character of the transition. This was implicitly illustrated in Fig. 6 of the original manuscript, where the nucleation of pearl in a domino phase is shown. No long-range diffusion was required in this simulation, which is already hinted at by the fast homogeneous nucleation (~ 1 ns) of the pearl phase. Since this point might not be very clear, though, we now additionally show this in Fig. R3 with an analysis of the atomic displacements during the pearl phase nucleation. Here, no long-range atomic displacements or diffusion is required. We clarified this by including the analysis from Fig. R3 in the supplemental as Fig. S7 and added the following clarifications to the updated text:

pg. 4, l. 206ff: “In any case where two GB phases have the same value of $[n]$, a phase transition can occur by local rearrangements of atoms and is thus considered to be diffusionless.”

pg. 6, l. 263ff: “An analysis of displacement vectors during the GB phase transition confirmed that only local atomic shuffling is required to create a pearl nucleus in the domino phase, showing that this GB phase transition is diffusionless (Supplementary Fig. S7).”

Past simulation results on grain boundary phase transitions did indeed report that diffusion is required for certain, specific transitions [Frolov et al., Structural phase transformations in metallic grain boundaries, Nat. Comm. 4, 1899 (2013)]. This is due to a difference in the excess number of atoms in the two grain boundary phases, expressed via the quantity $[n]$ in our manuscript. In the previous work on $\Sigma 19b$ grain boundaries [Meiners et al., Observations of grain-boundary

Figure R3: Minimal required displacements to transform domino into pearl. We produced two simulation cells, one with only the domino phase (top) and one where we inserted a pearl nucleus (middle). We then calculated the smallest displacements between these cells, displayed as arrows in the bottom panel. The simulations show that displacements on the order of 1 \AA are required, except for the B motifs, where two jumps of approximately 2 \AA take place. No long-range diffusion is necessary for this phase transition.

phase transformations in an elemental metal, Nature 579, 375 (2020)], as well as in the present work, all phases have $[n] = 0$ and therefore require no diffusion to transform into each other. The possibility of such a phenomenon is not too surprising, since diffusionless phase transformations are also well-known for bulk phases (e.g. the martensitic transformation). There is, of course, a temperature effect insofar that the equilibrium grain boundary phase changes at a transition temperature. This is not due to diffusion, however, but a consequence of a difference in entropy of the grain boundary phases ($-T[S]$ term of the excess free energy).

REVIEWERS' COMMENTS

Reviewer #1 (Remarks to the Author):

Thank you for responding to my comments.

Reviewer #2 (Remarks to the Author):

The authors have gone above and beyond in answering my comments. In my opinion the manuscript is ready for publication.

Reviewer #3 (Remarks to the Author):

This is a good study of the microstructure of grain boundaries. The authors nicely answered all my concerns and I recommend the paper for publication as it is.